# Asymmetry and changes in the neuromuscular profile of short-track athletes as a result of strength training

**Paweł Pakosz**[1]*, **Anna Lukanova-Jakubowska**[1], **Edyta Łuszczki**[2], **Mariusz Gnoiński**[3], **Oscar García-García**[4]

**1** Faculty of Physical Education and Physiotherapy, Opole University of Technology, Opole, Poland, **2** Institute of Health Sciences, Medical College of Rzeszow University, Rzeszow, Poland, **3** Pain Treatment Centre, Opole, Poland, **4** University of Vigo Pontevedra, Pontevedra, Spain

* p.pakosz@po.edu.pl

## Abstract

### Background

The purpose of this study was to identify the biomedical signals of short-track athletes by evaluating the effects of monthly strength training on changes in their neuromuscular profile, strength, and power parameters of the lower limb muscles. Muscle asymmetry, which can cause a risk of injury, was also evaluated.

### Methods and results

This study involved female athletes, age 18.8 ± 2.7 years, with a height of 162 ± 2.4 cm, and weight of 55.9 ± 3.9 kg. Before and after the monthly preparatory period prior to the season, strength measurements were assessed through the Swift SpeedMat platform, and reactivity of the lower limb muscles was assessed with tensiomyography (TMG). The athletes were also tested before and after the recovery training period. In the test after strength training, all average countermovement jump (CMJ) results improved. Flight time showed an increase with a moderate to large effect, using both legs (5.21%). Among the TMG parameters, time contraction (Tc) changed globally with a decrease (-5.20%). Changes in the results of the test after recovery training were most often not significant.

### Conclusion

A monthly period of strength training changes the neuromuscular profile of short-track female athletes, with no significant differences between the right and left lower limbs.

## Introduction

Short-track is a dynamic and speed sport discipline. Short-track is also a discipline that is becoming more popular, is unresearched, and offers great opportunities to improve athletes'

**Data Availability Statement:** All relevant data are within the paper and its Supporting Information files.

**Funding:** The authors received no specific funding for this work.

**Competing interests:** The authors have declared that no competing interests exist.

performance. One of the most important performance indicators for athletes in this discipline seems to be neuromuscular conduction and the related dynamics and speed of response of the muscles [1]. It is extremely important even at the start, where athletes want to achieve as high a rank as possible on the first bend. Then the whole race abounds in dynamic accelerations to obtain a higher rank or defend the present race. The race ends with a dynamic acceleration towards the finish [2]. On the other hand, speed skaters need to adopt a biomechanically favourable crouched position that is essential for the best skating speed performance. At the same time, high intramuscular forces lead to physiological disadvantages and fatigue. Therefore, short-track training is physiologically demanding and requires longer recovery periods than, for example, long track training [3, 4].

Importantly, the short-track dynamics of the muscles are often related to their strength. Both of these parameters are strongly correlated with each other; the greater the muscle strength, the faster the athlete [5]. (Jiménez-Reyes i inni, 2014) The dynamics are usually determined using invasive or effort-based testing methods based on parameters such as maximum strength, anatomical section area, maximum speed of turning without a load, or by testing power [6–8]. (Abe, Tayashiki, Nakatani i Watanabe, 2016; Ratamess i inni, 2016; Chelly, Fathloun, Cherif i Praagh, 2009) In the context of an assessment of specific muscles, tensiomyography (TMG) has been widely recognized, especially in recent years, as this method allows for the contraction properties of particular skeletal muscles to be noninvasively determined. This is a method of evaluating the biosignals emitted by the muscles as a result of electrical stimulation. It makes it possible to obtain quick and precise information on a muscle's characteristics, while not interfering with the athlete's training process [9–16]. (Atiković, Samardžija Pavletić i Tabaković, 2015) Using TMG, it is also possible to investigate muscle fibre status and a change in speed, strength or endurance in these muscles [17, 18].

This type of testing in short-track seems to be uncommon, and the need for data in this discipline is clear. On the other hand, in other sport disciplines, TMG measures top-level athletes controls neuromuscular conduction, and determines and restores correct reference values for the body's muscles. It is also a tool for objective screening tests in athletes, or to determine the percentage of particular types of myocytes in a muscle [19–22]. (Macgregor, 2016; García-García, Carral, Martínez-Trigo i Serrano-Gomez, 2013) (Simunič i inni, 2011; Pakosz, Jakubowska-Lukanova i Gnoiński, 2016)

The best short-track results depend to a considerable extent on the high driving force of the lower limb muscles generated by the athlete. It affects the acceleration and speeds obtained on ice. The muscles of the lower limb that contribute most to high speeds are the extensor muscles [23]. Since they are important muscles for races, the extensor muscles of both lower limbs were analysed in this study. Moreover, muscular asymmetry was examined because in short tracks, athletes always skate to the left, which could potentially create asymmetries that may cause injuries [21]. On the other hand, recent studies suggest that athletic asymmetries do not generally have a clear effect on athletic performance and that training interventions can also reduce athletic asymmetries [24].

The purpose of this study was to assess the strength and characteristics of short-track female athletes' muscles, as well as biomedical signal changes in the neuromuscular profile under the effect of a monthly regimen of exercises focused on improving muscle strength parameters. To check the effect of this training, a study was also performed of a monthly period of exercises focused on recovery, which took place before the strength training. The study also assessed the extent to which the muscle parameters of both lower limbs differed in athletes who practiced this asymmetrical sport. Reference values for this discipline were also determined. The research hypotheses assumed that there is a change in biomedical signals after one month of training aimed at increasing strength in athletes, which does not occur in recovery training.

More specifically, in the TMG-measured parameters responsible for muscle dynamics, Tc (contraction time), Dm (maximal displacement), and Td (delay time) will decrease after strength training. This will occur with a simultaneous increase in countermovement jump parameters of the lower limbs, measured with a contact mat. Strength training was also expected to influence lower limb asymmetry, which may affect the risk of injury.

## Materials and methods

### Subjects

The tests were carried out on the seven best female athletes from the Polish national short-track team (age 18.8 ± 2.7 years, body height 162 ± 2.4 cm, and body mass 55.9 ± 3.9 kg), who had no previous injuries in the measured muscles. To make the results homogeneous, the study was conducted only on athletes who had been training with the same training system for 4 years. This is a group of international female athletes, who have won medals in World Cups, World Championships, and European Championships. These athletes also repeatedly improved Polish national records for each distance. All participants were informed about the potential risk related to the examination and were informed about the purpose and course of the tests. They also signed an informed consent form confirming permission to participate in the tests, approved by the Bioethics Commission of the Chamber of Physicians in Opole No. 260, following the guidelines specified in the Declaration of Helsinki on human experimentation. In the case of underage persons, informed consent to participate in the tests was signed by the parents.

### Study design

A pretest-posttest quasi-experimental intrasubject design was used. Two measurements were conducted with the Swift SpeedMat platform and with the TMG system before and after the monthly recovery period (test after recovery training) and before and after the preparation period for the competition season (test after strength training). A measurement of the effects of regeneration training took place before the preparatory training tests, according to the athletes' annual macrocycle.

The monthly recovery training period (detraining) takes place before the preparation training period. It consists of one introductory microcycle and three regenerative microcycles (Table 1). Training took place 4–6 times per microcycle. The main training was the LT threshold training performed in the form of running or cycling at a set heart rate using the continuous method. Such training was performed by the athletes 2–3 times per microcycle. Strength training took place once a week during the recovery period. Strength endurance was performed in 2–3 series, with 12–15 repetitions. The main exercises were squats; forward, sideways and backward lunges; exercises to strengthen the core muscles; stability and balance exercises and strengthening of deep muscles with rubber bands. Secondary training during this period was supplementary suited to the individual needs of each athlete separately. Once per microcycle, the athletes played several games combined with flexibility training.

Table 1. Number of training units (in microcycles) in the recovery training period.

| Microcycle | Aerobic endurance—LT threshold training | Strength/Power | Supportive training outside the ice |
|---|---|---|---|
| Introductory | 2 | 1 | - |
| Regenerative | 3 | 1 | 2 |
| Regenerative | 3 | 1 | 2 |
| Regenerative | 3 | 1 | 2 |

**Table 2. Number of training units in microcycles in the preparation training period.**

| Microcycle | Aerobic endurance—LT threshold training | Strength/Power | Anaerobic training | Supportive training on ice |
|---|---|---|---|---|
| Introductory | 4 | 2 | 1 | 4 |
| Building | 3 | 3 | 2 | 3 |
| Building | 3 | 3 | 2 | 3 |
| Regenerative | 3 | 1 | - | 2 |

The monthly preparation training period consisted of 4 microcycles of 7 days (one introductory, two-building, and one regenerative) (Table 2). The number of training units in the introductory and building microcycles was 11, and there were 6 units in the regenerative microcycle. Training time per day was from 4 to 6 hours, and 1 to 3 hours in the regenerative microcycle. Training took place on the ice, on the athletic track, at the gym, and on bicycles.

The first direction of training was to increase lactate tolerance (training in the anaerobic zone). Basic training took place twice in the building microcycle (usually Tuesdays and Saturdays). Training included an ice ride during the time period from 50 s to 105 s at 90% speed, with a 60-s gap between segments. In two or three series of training, the number of repetitions was 7–10. This training causes a very high accumulation of lactate in the blood of athletes. For this reason, the training plan also included typical aerobic training, such as riding a bike or running. The second and main training was strength training. It took place three times in the building microcycle and twice in the introductory microcycle. In the regenerating microcycle, one training unit was used for muscle stimulation. Strength training was aimed at maintaining maximum strength and building maximum power. The main exercise was squats with a barbell. In the training unit, the athletes performed 2 x 6 repetitions of squats with a 70% max load, 2 x 4 repetitions with an 85% max load, 2 x 2 repetitions with a 90% max load, and 1 x 1 repetition with a 100% load. The load was controlled and corrected every three weeks. Power training took place as various types of jumps (through hurdles, stairs, or various types of obstacles), barbell jumps (from semi-squat—4 x 10 jumps with a 60% of max load, from full squat—4 x 6 jumps with a 50% of max load), sprints with loads and heavy ball throws. These exercises were selected individually depending on the athletes' needs. Complementary training was speed riding on ice, training in ice-riding techniques, and tactics training. This training was conducted by an individual or relay race. Such training units had a connecting role between the main training and the support training.

To report the intraday reliability of the measurement (Swift SpeedMat platform and TMG) for each evaluation, two measurements were performed in each athlete separated by a period of 15 min. All measurements were carried out by the same evaluator who had extensive experience in the use of both tools.

## Procedures

A contact platform (Swift SpeedMat, Wacoi, QLD, Australia) was used, with which the countermovement jump parameters for both feet and each of the lower limbs were measured separately. The tested subjects performed three jumps in three different protocols of the maximum vertical jump (CMJ–countermovement jump): 1) on the right foot, 2) on the left foot, and 3) on both feet. In total, each athlete made 9 jumps and had an approximately 30 s pause before each attempt. The highest jump from each attempt was selected for analysis. The jump started from the vertical position, standing with the hands on the hips, then a down movement was made by bending the knees and the hips, and then immediately straightening the knees and the hips jumping vertically up from the ground, to end by landing on both feet. During the

warm-up, athletes became familiar with the CMJ techniques, and these jumps were not taken for the calculation. After the athletes had learned the correct jumping technique, the examination started. The athletes entered the platform by jumping as high as possible. Participants started their attempts with a jump on the right foot and saw their jump height. The attempt was excluded and conducted again if at least one of the following elements occurred: poor technique, less than full contact of the foot with the platform, incomplete effort, or poor landing. Flight time (FT) in seconds, jump height (JH) in metres, jump power (JP) in watts, and relative power (W·kg$^{-1}$) were measured during each countermovement jump. The contact platform made the measurements with a time accuracy of 0.001 s.

For the test with the TMG system, data from the following muscles of the right and left lower limbs of the short-track female athletes were used: musculus gastrocnemius caput mediale (GM), musculus gastrocnemius caput laterale (GL), musculus tibialis anterior (TA), musculus vastus medialis (VM), musculus vastus lateralis (VL), musculus rectus femoris (RF), musculus gluteus maximus (GT), and musculus biceps femoris (BF). While examining the muscles, the following parameters were measured: maximum radial muscle belly displacement (Dm), in mm. Contraction time (Tc) is the time in ms from 10 to 90% of Dm (see Fig 1). Delay time (Td) as the time, in ms, from the onset to 10% of Dm; and V$_{90}$ as the rate (mm·s$^{-1}$) between the radial displacement occurring during the period of Tc + Td (Dm90) and Tc + Td [Dm90/Tc + Td]. Measurements were carried out with the examined subjects after they assumed supine or prone decubitus position. The right angle in the joint was maintained with the help of a triangle foam pad supporting the leg. The extensor muscles of the knee were measured in the knee joint setting at an angle of 120˚, while the flexor muscles of the knee were measured at an angle of 150˚.

The signal-collecting pressure sensor was connected to a precise digital displacement converter and fixed perpendicular to the muscle belly. The digital displacement converter (GK 40 Panoptik d.o.o., Ljubljana, Slovenia) had a 0.17 N/mm spring installed. The sensor had a controlled initial pressure of 1.5 x 10$^{-2}$ N/mm$^2$. During measurement, as a result of the electric stimulation of the muscles, the displacement sensor is pressed against the skin, and the measurement results are presented in the form of time and displacement curves. The sensor was set perpendicular to the thickest part of the muscle belly. The thickest part of the muscle belly was determined visually and through palpation during a voluntary contraction. The muscle was stimulated by two self-adhesive electrodes (Axelgaard, Pulse) placed 2–5 cm from one another, invoking a 1-millisecond impulse from the electrostimulator (TMG-S1, Furlan and Co. ltd.). The diameter of the electrodes and their placement were selected based on the size of

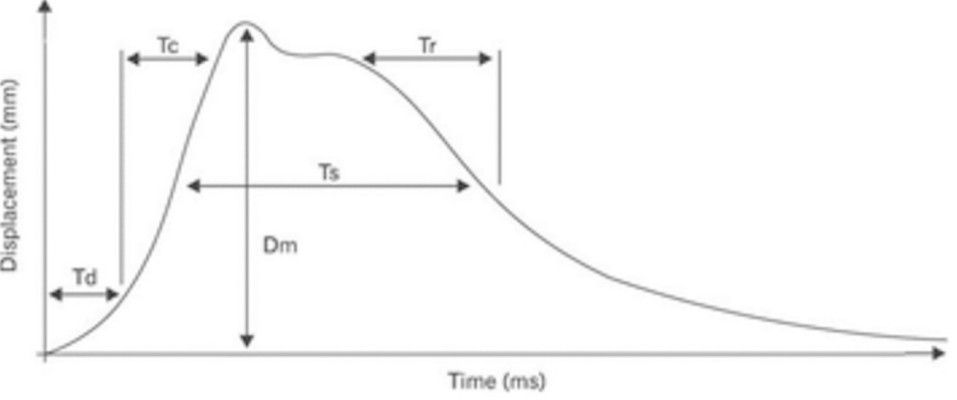

**Fig 1. TMG displacement curve along with the parameters.**

the muscles to isolate the contraction of the particular muscle, and to avoid simultaneous activation of the nearby muscles. (Rodríguez-Matoso i inni, 2010)Electrical stimulation was applied with a pulse duration of 1 ms and an initial current amplitude of 30 mA, which was progressively increased in 10 mA steps until reaching 100 mA (maximal stimulator output). Ten-second intervals were maintained between each impulse. The digital TMG signal was taken directly from the MATLAB Compiler Toolbox using a 1-kHz sampling frequency. The TMG signal was saved and stored on a portable PC. The maximum amplitude of the stimulation was recognized as the minimum amplitude needed for a response with the highest displacement of the muscle (Dm).

## Statistical analyses

Relative reliability was calculated through intraclass correlation coefficient (ICC) analysis using a single measurement, 2-way mixed-effects model, and absolute agreement. Generally, a value less than 0.5 is considered an indicator of poor reliability, values between 0.5 and 0.75 indicate moderate reliability, values between 0.75 and 0.9 indicate good reliability, and values greater than 0.9 indicate excellent reliability [25]. The coefficient of variation (CV) was used as a measure of absolute reproducibility [26]. The percentage of standard error of measurement (%SEM) has also been used as an absolute reliability measure. SEM = $\sqrt{\text{MSE}}$, where MSE is the mean square error term from the repeated-measures ANOVA. %SEM was calculated as SEM/M × 100, where M is the mean of the two intraday measurements. To detect significant differences in the measurements after the monthly period of training, the Kruskal-Wallis test was used taking as factors time (before vs. after), side (right vs. left), and muscle. The effect size of the percentage differences was calculated using Cohen's d, with values of 0.2, 0.5, and 0.8 used to represent small, moderate, and large differences, respectively [27]. To relate the variables of TMG with the jump performance variables, Pearson's bivariate correlation coefficient was used. An alpha level of $p < .05$ was considered statistically significant. All data were analysed using SPSS v21.0 for Windows (SPSS Inc., Chicago, IL, USA). In relation to the topic of this paper, the paper first presents the results of a strength training study.

## Results

The reliability values (ICC, 95% CI; CV and %SEM) obtained were: FT .99 (.95-.99), cv 1%, SEM .96%; JH .99 (.94-.99), cv 2.1%, SEM 1.94%; JP .99 (.97-.99), cv 2.1%, 1.87%; Dm .92 (.80-.97), cv 6.5%, SEM 7.35%; Tc .92 (.80 - .96), cv 4.4%, SEM 4.37%; Td .93 (.84 - .97), cv 3.4%, SEM 2.89%. After the monthly strength training period, the parameters of all average countermovement jump results improved, p = 0.003 (Table 3). No difference was found between the left leg and the right leg (p = 0.846). Flight time showed an increase with a moderate-large effect size, with both legs (5.21%; p = 0.02; ES = 0.67) and with the right (6.47%; p = 0.04; ES = 0.9) and left legs (7.28%; p = 0.02; ES = 1). The CMJ variables jump height, jump power, and W·kg$^{-1}$ showed the same trend. However, the effect size of the power parameters is small for both the right and left legs.

On the other hand, after the monthly recovery training period, the parameters of all average countermovement jump results also improved, p = 0.342, with no difference between the right and left legs, p = 0.792 (Table 4). Only the flight height at the left leg changed significantly in the second measurement (9.52%; p = 0.03; ES = 1).

No difference was found between the left leg and the right leg in terms of TMG parameters during the strength training period (Table 5). After the monthly training period, only Tc changed globally with a small decrease (29.41 ± 12.83 vs. 27.88 ± 11.67 ms, -5.20%; p = 0.02; ES = 0.13). There were significant differences (p = 0.0001) between the muscles in all

**Table 3. Results of the countermovement jump measurement on the Swift SpeedMat platform before and after the monthly strength training period.**

| CMJ Parameter | Side | Before | After | % Difference | ES |
|---|---|---|---|---|---|
| Flight Time both feet | Both | 0.520 ± 0.03 | 0.547 ± 0.04 | 5.21* | 0.67 |
| Flight Time one foot | Right | 0.417 ± 0.04 | 0.444 ± 0.03 | 6.47* | 0.90 |
| | Left | 0.412 ± 0.03 | 0.442 ± 0.03 | 7.28* | 1 |
| Height both feet | Both | 0.33 ± 0.04 | 0.37 ± 0.06 | 12.12* | 0.66 |
| Height one foot | Right | 0.214 ± 0.04 | 0.244 ± 0.03 | 14.01* | 1 |
| | Left | 0.208 ± 0.04 | 0.242 ± 0.03 | 16.34* | 1.13 |
| Power both feet | Both | 735.49 ± 307.42 | 810.2 ± 326.46 | 10.15* | 0.22 |
| Power one foot | Right | 488.07 ± 324.44 | 549.93 ± 290.03 | 12.67* | 0.21 |
| | Left | 474.74 ± 298.14 | 545.51 ± 287.37 | 14.90* | 0.24 |
| $W \cdot kg^{-1}$ | Both | 12.43 ± 3.82 | 13.82 ± 4.24 | 11.18* | 0.33 |

Flight time was measured in seconds: height in metres; power in watts.

*Statistical significance of the changes at the level of $p < 0.05$, compared to the first measurement.

parameters of the TMG; for example, GT was the one with the highest values of Tc, Td, and Dm obtained, and VM was the one that presented most $V_{90}$.

Analysing particular muscle groups separately, the Tc of GL, GM, RF, and VM decreased with a small-to-large effect size; however, the Tc of GT increased with a large effect size. The Td of VM decreased with a moderate effect size. The Dm of GT decreased with large effect size, and the $V_{90}$ of TA increased moderately.

A low positive correlation was found between flight time (before and after) and the Dm and $V_{90}$ of muscle evaluations. Dm before ($r = .32$; $p = .001$), Dm after ($r = .26$; $p = .004$), $V_{90}$ ($r = .37$; $p = .001$) and $V_{90}$ ($r = .35$; $p = .001$).

Significantly fewer differences between the two measurements were seen after recovery training (Table 6). Globally, no parameter significantly changed after this monthly training period. For all TMG parameters, there were significant differences between muscles ($p = 0.001$).

After analysing the particular muscle groups separately, it was determined that the Tc of TA decreased with a large effect size, and the Tc of GT increased with a large effect size. The Td of RF decreased with a large effect size. On the other hand, Dm and $V_{90}$ showed no significant changes. No correlation was found between parameters.

**Table 4. Results of the countermovement jump measurement on the Swift SpeedMat platform before and after the monthly recovery period.**

| CMJ Parameter | Side | Before | After | % Difference | ES |
|---|---|---|---|---|---|
| Flight Time both feet | Both | 0.535 ± 0.03 | 0.537 ± 0.02 | 0,37 | 0,10 |
| Flight Time one foot | Right | 0.422 ± 0.02 | 0.438 ± 0.04 | 3,79 | 0,40 |
| | Left | 0.419 ± 0.02 | 0.435 ± 0.03 | 3,82 | 0,53 |
| Height both feet | Both | 0.35 ± 0.03 | 0.36 ± 0.03 | 2,86 | 0,33 |
| Height one foot | Right | 0.23 ± 0.03 | 0.24 ± 0.04 | 4,35 | 0,25 |
| | Left | 0.21 ± 0.03 | 0.23 ± 0.02 | 9,52* | 1,00 |
| Power both feet | Both | 808,1 ± 289.1 | 809.6 ± 280.7 | 0,19 | 0,01 |
| Power one foot | Right | 498,5 ± 254.5 | 510.7 ± 255.1 | 2,45 | 0,05 |
| | Left | 491,2 ± 246,2 | 505.3 ± 268.3 | 2,87 | 0,05 |
| $W \cdot kg^{-1}$ | Both | 13.1 ± 3.7 | 13.4 ± 4.1 | 2,29 | 0,07 |

Flight time was measured in seconds: height in metres; power in watts.

*Statistical significance of the changes at the level of $p < 0.05$, compared to the first measurement.

**Table 5. Measurement results of TMG parameters for specific muscles before and after the monthly strength training period.**

| Measured TMG parameter | m. BF | m. GL | m. GM | m. GT | m. RF | m. VL | m. VM | m. TA |
|---|---|---|---|---|---|---|---|---|
| Tc before | 30.73 ± 18.62 | 37.27 ± 20.83 | 25.11 ± 3.66 | 46.47 ± 4.48 | 27.62 ± 4.09 | 22.75 ± 2.82 | 24.97 ± 1.68 | 20.35 ± 1.71 |
| Tc after | 32.79 ± 11.99 | 23.46 ± 2.42* | 22.27 ± 2.02* | 53.56 ± 5.64* | 25.37 ± 5.41* | 22.25 ± 2.72 | 22.84 ± 1.68* | 21.54 ± 9.47 |
| % difference and ES | 6.70% 0.17 | -37.05% 5.6 | -11.31% 1.40 | 15.25% 1.25 | -8.14% 0.41 | -2.19% 0.18 | -8.53% 1.26 | 5.84% 0.12 |
| Td before | 22.82 ± 2.90 | 21.29 ± 2.07 | 20.85 ± 1.23 | 31.52 ± 3.36 | 24.09 ± 1.24 | 21.73 ± 1.20 | 23.04 ± 1.04 | 21.43 ± 1.56 |
| Td after | 23.96 ± 2.20 | 20.26 ± 0.98 | 20.49 ± 1.26 | 32.99 ± 2.73 | 23.79 ± 1.08 | 21.33 ± 1.28 | 22.11 ± 1.31* | 21.22 ± 1.66 |
| % difference and ES | 4.99% 0.51 | -4.83% 1.05 | -1.72% 0.28 | 4.66% 0.53 | -1.24% 0.11 | -1.84% 0.31 | -4.03% 0.70 | -0.97% 0.12 |
| Dm before | 4.47 ± 3.09 | 5.04 ± 2.04 | 3.54 ± 1.01 | 9.60 ± 4.56 | 6.64 ± 1.76 | 5.81 ± 1.54 | 7.78 ± 1.07 | 2.53 ± 0.75 |
| Dm after | 4.99 ± 2.88 | 3.81 ± 0.83* | 3.16 ± 0.58 | 10.92 ± 4.21 | 6.36 ± 2.12 | 5.38 ± 1.23 | 7.58 ± 1.35 | 3.12 ± 1.14 |
| % difference and ES | 11.63% 0.18 | -24.40% 1.48 | -10.73% 0.65 | 13.75% 0.31 | -4.21% 0.13 | -7.40% 0.34 | -2.57% 0.14 | 23.32% 0.51 |
| V$_{90}$ before | 71.50±41.74 | 81.32±26.58 | 69.23±17.34 | 111.39±55.22 | 115.58±29.59 | 116.73±26.28 | 146.15±21.09 | 54.62±16.74 |
| V$_{90}$ after | 74.20±30.60 | 78.90±17.21 | 67.01±13.76 | 118.08±56.34 | 116.66±38.18 | 111.12±24.44 | 151.60±24.85 | 65.74±19.61* |
| % difference and ES | 3.77% 0.08 | -2.97% 0.14 | -3.20% 0.16 | 6% 0.11 | 0.93% 0.02 | -4.80% 0.22 | 3.72% 0.21 | 20.35% 0.56 |

## Discussion

The main findings were that, after performing a monthly training period aimed at increasing maximum strength, the parameters of all average countermovement jump results improved, and muscle contraction time slightly decreased. Furthermore, the Dm and V$_{90}$ of the muscles evaluated slightly correlated positively with flight time in both the previous and post training assessments.

In our research, the ICC and CV values obtained were good to excellent for all parameter evaluations, which corroborates the good reproducibility of the evaluated TMG parameters, such as those obtained by Martín-Rodríguez et al. [28], and the countermovement jump measurement, such as that obtained by Jiménez-Reyes et al. [29].

After the monthly strength training program, differences in the strength of the lower limb muscles, measured according to contact mat parameters—flight time, flight height, and power—were demonstrated in the research we conducted. Therefore, we confirmed the hypothesis that strength training contributes to a significant improvement in the height, time, and power of the jumps. Such results are confirmed by many studies [30–33]. Moreover, the greatest force effect after the training period was evident in each of the three jumping cases during testing, which is similar to what was found by Pritchard et al. [34], where flight time also increased

**Table 6. Measurement results of TMG parameters for specific muscles before and after the monthly recovery training period.**

| Measured TMG parameter | m. BF | m. GL | m. GM | m. GT | m. RF | m. VL | m. VM | m. TA |
|---|---|---|---|---|---|---|---|---|
| Tc before | 37.38 ± 19.66 | 30.49 ± 17.80 | 21.57 ± 3.20 | 47.49 ± 5.09 | 28.64 ± 3.98 | 23.36 ± 2.75 | 26.51 ± 1.71 | 35.77 ± 3.44 |
| Tc after | 43.06 ± 13.45 | 26.61 ± 17.1 | 20.55 ± 1.91 | 59.60 ± 5.72* | 28.74 ± 4.23 | 21.18 ± 2.34 | 25.49 ± 1.42 | 26.60 ± 4.91* |
| % difference and ES | 15.17% 0.42 | -12.71% 0.23 | -4.75% 0.54 | 25.51% 2.12 | 0.36% 0.02 | -9.34% 0.93 | -3.83% 0.71 | -25,65% 1.87 |
| Td before | 24.40 ± 3.10 | 20.43 ± 1.98 | 20.17 ± 1.92 | 33.26 ± 3.91 | 25.19 ± 1.82 | 22.32 ± 1.13 | 21.74 ± 1.20 | 23.61 ± 1.82 |
| Td after | 25.06 ± 3.20 | 20.67 ± 1.20 | 19.17 ± 1.34 | 35.56 ± 3.97 | 23.58 ± 1.23* | 21.69 ± 1.27 | 21.89 ± 1.48 | 22.42 ± 1.36 |
| % difference and ES | 2.66% 0.20 | -1.19% 0.20 | -4.98% 0.75 | 6.91% 0.58 | -6.40% 1.31 | -2.84% 0.50 | 0.67% 0.10 | -5.03% 0.87 |
| Dm before | 5,98 ± 2.69 | 4.00 ± 2.74 | 2.53± 1.32 | 7.95 ± 4.62 | 7.08 ± 1.78 | 4.89 ± 1.46 | 6.25 ± 1.58 | 3.78 ± 0.56 |
| Dm after | 6.86 ± 2.91 | 4.42 ± 0.93 | 2.24 ± 1.49 | 9.33 ± 4.48 | 7.12 ± 1.92 | 5.25 ± 1.65 | 7.02 ± 0.87 | 2.94 ± 1.23 |
| % difference and ES | 14.77% 0.30 | 10.36% 0.45 | -11.59% 0.20 | 17.40% 0.31 | 0.58% 0.02 | 7.29% 0.22 | 12.42% 0.89 | -22.42% 0.69 |
| V$_{90}$ before | 87.06±42.40 | 70.76±27.10 | 54.55±15.30 | 88.56±50.23 | 118.35±30.93 | 96.33±24.30 | 116.54±25.32 | 57.34±17.60 |
| V$_{90}$ after | 90.64±35.90 | 84.09±19.20 | 50.69±16.10 | 88.23±52.31 | 122.47±36.72 | 110.14±23.54 | 133.43±23.49 | 53.89±19.21 |
| % difference and ES | 4.12% 0.10 | 18.83% 0.69 | -7.07% 0.24 | -0.38% 0.01 | 3.48% 0.11 | 14.34% 0.59 | 14.49% 0.72 | -6.02% 0.18 |

significantly after 4-week strength training. On the other hand, no significant changes were found after recovery training, such as after strength training.

Scientific research on skaters has proven that in short-track, the muscles of the right lower limb are more loaded when skating, which is related to the specific character of this discipline. This was determined using EMG [23, 35, 36], for example, and after checking the desaturation (reduction in blood oxygen-saturation) of the extensor muscles of the thigh [37]. When training in this discipline, it is extremely difficult to balance the work of both lower limbs. However, athletes spend a significant part of their training on ice, where they skate only to the left, and this, in turn, results in a higher load on the right limb. However, the present research has confirmed that symmetry of the lower limbs under the influence of training is possible. The small visible differences between the leg muscles are within the limits of statistical error. This may indicate a well-performed training process, and the symmetry found in the studies may cause fewer injuries [21]. In research, by examining changes in TMG parameters, it was also determined how different muscles respond to monthly strength training, which was a preparatory period to the season. (Lehnert, Psotta i Botek, 2012)

TMG parameter analyses have confirmed significant differences in lower limbs as a result of strength training. Indeed, the muscle properties studied with TMG are sensitive to changes in muscle force, which is also indicated by the results of De Paula Simola et al. [18]. (Simola i inni, 2015) When measuring the muscles with TMG, the Tc parameter was significantly decreased. These results confirm that changes occur in the neuromuscular profile of the lower limbs as a result of training, as determined by García-García et al. [38]. (García-García, Cancela-Carra i Huelin-Trillo, Neuromuscular profile of top-level women kayakers assessed through tensiomyography, 2014) The highest decrease in the Tc parameter was observed in plantar flexion muscles, GL and GM, and knee extensor muscles, RF, VL, and VM. The tendency to decrease the parameter value in these muscles can also be observed in parameters Td, Dm, and $V_{90}$, but these are usually not statistically significant changes. The above muscles are responsible for the moment of a rebound during the jump and skate from the ice during skating, and they contribute most to short-track high speeds [23]. The remaining examined muscles usually had an increasing tendency of TMG parameters. On the other hand, after monthly recovery training, no significant changes were found.

The remaining TMG parameters, Td, Dm, and $V_{90}$, had a smaller tendency to change during the second study. Particularly surprising is the lack of significant changes in the Dm parameter, which is often one of the most significant TMG parameters in studies. According to TMG research, the lower results of the Dm parameter indicate a good predisposition to high-strength and dynamic tasks such as sprints and jumps [12, 18], which are also important in short-track, and the monthly training period is supposed to increase strength.

When examining whether there were correlations between the examined parameters, a low positive correlation was detected between flight time and Dm and $V_{90}$ before and after the training period. In this sense, TMG has been used to establish relationships between neuromuscular parameters and sport performance indicators [39]. The TMG parameters have also been related to jumping performance. Nevertheless, this relationship is unclear. On the one hand, power athletes' highest performance in jumping tests is related to lower values of Tc, Dm, and Td in RF and BF [40], but on the other hand, Gil et al. [11] found no correlations between TMG parameters and height of jumps and sprint velocity in professional soccer players. The present results indicate a logically low positive correlation between flight time and $V_{90}$, but the relationship with Dm is not clear from the point of view of muscle-tendon stiffness, so more research is still needed on the relationship that may exist between contractile properties and jumping performance.

An analysis of TMG parameters, as well as in the case of SpeedMat, showed no significant differences between the right and left lower limbs, which is consistent with the study by Gil et al. [11] concerning the similar performance found between the lower limbs in athletes. (Gil S. i inni, 2015)However, differences between the sides were found; after one month of strength training, they decreased. This may therefore indicate that this strength training reduces asymmetries, which is confirmed by the Maloney study [24].

The limitation of this study was the small number of examined female athletes. However, a coherent group of athletes who had uniform training and the same coach during that time were examined. All the basic surface muscles of the lower extremities were also examined. As a result, data were obtained from athletes having similar daily stimuli from a very uniform group. This type of research is also innovative and aims to show future possible trends in muscle strength research.

## Conclusions

Using TMG, the reference values of the lower limb muscles were determined for the top-level short-track female athletes. The data for neuromuscular conduction of the extensor muscles in the lower limbs may turn out to be particularly significant, as these muscles have a decisive impact on the rebound of the lower limbs against the ice pane. In research, these muscles proved their strong effect on specialist formation and improvement in the jumping ability, power, and TMG parameters. By monitoring the TMG parameters, a coach could also obtain more information on how to individualize loads for athletes while simultaneously controlling the training effects in the annual training cycle. In addition, TMG examination is noninvasive, does not result in fatigue, and does not affect the load structure of training. A better understanding of the nature of muscle work in sports, examined with TMG on the example of short-track, may help coaches, athletes, doctors, physiotherapists, and scientists contribute to better sports performance achieved by their athletes or to prevent and treat their injuries. Therefore, methods that will help improve such processes should be continuously sought, and one of them is certainly tensiomyography (TMG).

After a one-month training period aimed at increasing maximum strength, the parameters of all average countermovement jump results improved, and muscle contraction time slightly decreased among the top-level short-track female athletes. In the test after recovery training, changes in results were most often not significant. Moreover, no significant differences between the right and left lower limbs were found in either research tool. The applied training process does not cause leg asymmetry in short-track athletes and thus prevents the risk of injury. TMG parameters Dm and $V_{90}$ of evaluated muscles slightly correlate positively with flight time in both the previous and post-strength training assessments.

## Supporting information

**S1 Dataset.**
(PDF)

**S2 Dataset.**
(PDF)

## Author Contributions

**Conceptualization:** Anna Lukanova-Jakubowska, Oscar García-García.

**Data curation:** Anna Lukanova-Jakubowska, Oscar García-García.

**Formal analysis:** Paweł Pakosz, Oscar García-García.

**Funding acquisition:** Paweł Pakosz.

**Investigation:** Paweł Pakosz, Mariusz Gnoiński.

**Methodology:** Paweł Pakosz, Anna Lukanova-Jakubowska, Edyta Łuszczki, Mariusz Gnoiński, Oscar García-García.

**Project administration:** Paweł Pakosz, Oscar García-García.

**Software:** Mariusz Gnoiński.

**Supervision:** Paweł Pakosz.

**Writing – original draft:** Paweł Pakosz, Edyta Łuszczki, Oscar García-García.

**Writing – review & editing:** Paweł Pakosz, Oscar García-García.

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
