## [Decision Letter · Decision Letter 0]

1 Sep 2021

PONE-D-21-22288

Asymmetry and changes in the neuromuscular profile as a result of strength training

PLOS ONE

Dear Dr. Pakosz,

Thank you for submitting your manuscript to PLOS ONE. After careful consideration, we feel that it has merit but does not fully meet PLOS ONE’s publication criteria as it currently stands. Therefore, we invite you to submit a revised version of the manuscript that addresses the points raised during the review process.

Your ms has been revised by two experts in the field. As you can see in the report, there are several major points the  reviewers have raised. In particular, consider the indications provided by Reviewer 1 about the static approach used.  Also, consider making as strong revision of the English as both Reviewers indicated to revise it. Please submit your revised manuscript by Oct 16 2021 11:59PM. If you will need more time than this to complete your revisions, please reply to this message or contact the journal office at plosone@plos.org. Please include the following items when submitting your revised manuscript:

We look forward to receiving your revised manuscript.

Kind regards,

Emiliano Cè

Academic Editor

PLOS ONE

“No”

“NO authors have competing interests”

Reviewers' comments:

Reviewer's Responses to Questions

**Comments to the Author**

1. Is the manuscript technically sound, and do the data support the conclusions?

Reviewer #1: Yes

Reviewer #2: No

2. Has the statistical analysis been performed appropriately and rigorously? 

Reviewer #1: Yes

Reviewer #2: No

3. Have the authors made all data underlying the findings in their manuscript fully available?

Reviewer #1: Yes

Reviewer #2: Yes

4. Is the manuscript presented in an intelligible fashion and written in standard English?

Reviewer #1: Yes

Reviewer #2: No

5. Review Comments to the Author

Reviewer #1: The article is original and presents novel information in what concerns to short track athletes adaptations to strength training. The procedures are appropriate to accomplish the objectives of the study.

The English is understandable but should be improved.

The title is too generic. I recommend to change it to “Asymmetry and changes in the neuromuscular profile of short track athletes as a result of strength training”

In the introduction it would be important a briefly presentation of the physiological demands of the sport, and a brief overview of the TMG signals adaptations to strength training.

The sample size is small, but the fact that the participants are high level athletes in a poorly studied sport is a valuable input of the article.

Although the athletes are clearly very experienced in what concerns to the sport, their experience in what concerns to strength training may not be the same. With an average age of 18,8 years old and probably encompassing athletes u-18, the number of years of strength training may not be the same and this would result in different neuromuscular chronic adaptations of the athletes and therefore it would influence the magnitude of the results obtained in each athlete. Can you provide this information? Or can you ensure the homogeneity of the strength training status of the athletes?

The type, intensity and volume are described in the text, but it would be beneficial to the reader if this information is presented in a table. For example, a table with the microcycles of both mesocycles in lines and the training types in columns (endurance, strength/power and ice). It is optional to the author but I think it would help readers without knowledge about the sport, the physiological demands and the type of training.

It is not clear if the recovery mesocycle took place right after or before the preparation mesocycle.

The paragraph in ll.105-119 presents many repeated ideas of the previous paragraph. The sentences that are entirely equal should be rewritten or they should mention the above paragraph information.

The participants had a familiarization during the warm-up with the CMJ? It is a technical jump, and the learning during the first attempts should be considered.

ll.226. You start the sentence with “the differences in strength”. This should be used with caution because you do not measure strength of the lower limb directly in the jumping movement or indirectly in any maximum strength test (e.g. squat, leg press or mid-thigh pull), you measure the flight time and you assume that the observed differences are due to increased strength of the lower limbs. Am I right? I think this paragraph should be rewritten, and maybe you should describe information of other studies clearly showing that differences in flight time are caused by differences in peak force or rate of force development.

ll.247. It should be plantar flexion muscles GL and GM

Reviewer #2: The present study aimed to "assess the strength and characteristics of short track female athletes' muscles, as

well as biomedical signal changes in the neuromuscular profile under the effect of a monthly period of exercises focused on improvement in muscle strength parameters, and secondly focused on recovery."

I'd like to provide some positive feedback to the authors, although I have several major concerns. i) assessing elite athletes is always hard; ii) the detailed description of the training program and "recovery" program is not so common to read in scientific literature and it should be exactly the way the authors did it.

Concerns:

1. The manuscript is hard to read and follow the main ideas. I strongly recommend that authors some proof-reading. This is critical since the motivation and rationale is very confusing, and partly due to English issues.

2. The introduction hardly leads the reader to the proposed aim and hypotheses. Just to provide an example of what I mean. in the last paragraph of the intro one can read "In the short track, the athletes always

skate to the left, so it was also checked, how much the parameters of the muscles in both lower limbs vary among themselves in the athletes in this asymmetric sport discipline and to determine the reference values for the discipline."...which I believe the authors intended to discuss and point out potential asymmetries in this sport. However, such topic is hardly discussed within the intro and a common reader hardly understands why does it matter (is it really "bad" to be asymmetric in such "a asymmetric sport"). I am not trying to say it does, or does not, matter. But this needs a clear discussion so the reader can get the whole picture of what and why the authors are investigated such thing. Note this is just one example of the lack of a strong rationale within the introduction

3. statistical analysis: are you really calculating ICC analysis (note: please report SEM values), correlations, and a 2 factors design with an N =7? I am sorry, but no matter how large the effect size can be...this is potentially tremendously biased. I fully understand this is an elite group, and maybe you only have access to 7 elite athletes. But there are many others statistical approaches to deal with such type of data set (e.g. single-subject analysis.

4. Study design: At a certain point I got very confused. The authors mentioned a pre vs post, left vs right, and experimental vs control (note: this last one does not seem the best terminology to be). I couldn't see this design being tested in your statistical approach, I may have missed some.

I do have some other minor comments but I would like to see these ones addressed before further detailed comments. Also, considering my comments I first need to be clarified in all these aspects before I comment the discussion and conclusion sections.

6. PLOS authors have the option to publish the peer review history of their article (what does this mean?). If published, this will include your full peer review and any attached files.

Reviewer #1: No

Reviewer #2: No

---

## [Author Response · Author response to Decision Letter 0]

21 Oct 2021

Reviewer #1: The article is original and presents novel information in what concerns to short track athletes adaptations to strength training. The procedures are appropriate to accomplish the objectives of the study.

The English is understandable but should be improved.

Thank you for your valuable comment and positive feedback. English language corrected, certificate attached. 

The title is too generic. I recommend to change it to “Asymmetry and changes in the neuromuscular profile of short track athletes as a result of strength training”

Thank you for your valuable comment. The title of the manuscript has been changed.

Asymmetry and changes in the neuromuscular profile of short-track athletes as a result of strength training

In the introduction it would be important a briefly presentation of the physiological demands of the sport, and a brief overview of the TMG signals adaptations to strength training.

We fully understand your comment. Added text to the manuscript as follows: 

On the other hand, speed skaters need to adopt a biomechanically favourable crouched position that is essential for the best skating performance. At the same time, high intramuscular forces lead to physiological disadvantages and fatigue. Therefore short-track training is physiologically demanding and requires longer recovery periods than for example long track training [3,4].

And

Using TMG, it is also possible to investigate muscle fibre status and a change in speed, strength or endurance in these muscles [17,18].

The sample size is small, but the fact that the participants are high level athletes in a poorly studied sport is a valuable input of the article.

Although the athletes are clearly very experienced in what concerns to the sport, their experience in what concerns to strength training may not be the same. With an average age of 18,8 years old and probably encompassing athletes u-18, the number of years of strength training may not be the same and this would result in different neuromuscular chronic adaptations of the athletes and therefore it would influence the magnitude of the results obtained in each athlete. Can you provide this information? Or can you ensure the homogeneity of the strength training status of the athletes?

Thank you for your valuable comment. The athletes were a selected national team, which had been preparing for 4 years with the same training cycle. Previously, they were trained by national sports associations with unified central training, so the amount of strength training was similar, even more so in the period immediately preceding the study. However, the text of the article was added for clarification purposes:

To make the results homogeneous, the study was conducted only on athletes who had been training with the same training system for 4 years.

The type, intensity and volume are described in the text, but it would be beneficial to the reader if this information is presented in a table. For example, a table with the microcycles of both mesocycles in lines and the training types in columns (endurance, strength/power and ice). It is optional to the author but I think it would help readers without knowledge about the sport, the physiological demands and the type of training.

Thank you for your valuable comment. This will actually make it easier for the reader to read, so Table 1 and 2 have been added to the text as suggested.

Table 1: Number of training units in microcycles in the recovery training period

Microcycle Aerobic endurance 

- LT threshold training Strength/Power

 Supportive training outside the ice

Introductory 2 1 -

Regenerative 3 1 2

Regenerative 3 1 2

Regenerative 3 1 2

Table 2: Number of training units in microcycles in the preparation training period

Microcycle Aerobic endurance - LT threshold training Strength/Power

 Anaerobic training

 Supportive training on ice

Introductory 4 2 1 4

Building 3 3 2 3

Building 3 3 2 3

Regenerative 3 1 - 2

It is not clear if the recovery mesocycle took place right after or before the preparation mesocycle.

We are grateful for your comment. Text added as follows:

A measurement of the effects of regeneration training took place before the preparatory training tests, according to the athletes’ annual macrocycle.

The paragraph in ll.105-119 presents many repeated ideas of the previous paragraph. The sentences that are entirely equal should be rewritten or they should mention the above paragraph information.

Thank you for your valuable comment. We have changed part of the manuscript as follows:

The monthly recovery training period (detraining) takes place before the preparation training period. It consists of one introductory microcycle and three regenerative microcycles (Table 1). Training took place 4-6 times per microcycle. The main training was the LT threshold training performed in the form of running or cycling at a set heart rate using the continuous method. Such training was performed by the athletes 2-3 times per microcycle. Strength training took place once a week during the recovery period. Strength endurance was performed in 2-3 series, with 12-15 repetitions . The main exercises were squats; forward, sideways and backward lunges; exercises to strengthen the core muscles; stability and balance exercises and strengthening of deep muscles with rubber bands. Secondary training during this period, was supplementary suited to the individual needs of each athlete separately. Once per microcycle, the athletes played several games combined with flexibility training.

The participants had a familiarization during the warm-up with the CMJ? It is a technical jump, and the learning during the first attempts should be considered.

We are grateful for your comment. Text added as follows:

During the warm-up, athletes became familiar with the CMJ techniques, and these jumps were not taken for the calculation. After the athletes had learned the correct jumping technique, the examination started.

ll.226. You start the sentence with “the differences in strength”. This should be used with caution because you do not measure strength of the lower limb directly in the jumping movement or indirectly in any maximum strength test (e.g. squat, leg press or mid-thigh pull), you measure the flight time and you assume that the observed differences are due to increased strength of the lower limbs. Am I right? I think this paragraph should be rewritten, and maybe you should describe information of other studies clearly showing that differences in flight time are caused by differences in peak force or rate of force development.

 We are grateful for your valuable comment. Reworded paragraph as follows:

After the monthly strength training program, differences in the strength of the lower limb muscles, measured according to contact mat parameters - flight time, flight height, and power - were demonstrated in the research we conducted. Therefore, we confirmed the hypothesis, that strength training contributes to a significant improvement in the height, time, and power of the jumps.

ll.247. It should be plantar flexion muscles GL and GM

Thank you for your valuable comment. Text changed as follows:

plantar flexion muscles, GL and GM,

 

Reviewer #2: The present study aimed to "assess the strength and characteristics of short track female athletes' muscles, as

well as biomedical signal changes in the neuromuscular profile under the effect of a monthly period of exercises focused on improvement in muscle strength parameters, and secondly focused on recovery."

I'd like to provide some positive feedback to the authors, although I have several major concerns. i) assessing elite athletes is always hard; ii) the detailed description of the training program and "recovery" program is not so common to read in scientific literature and it should be exactly the way the authors did it.

Concerns:

1. The manuscript is hard to read and follow the main ideas. I strongly recommend that authors some proof-reading. This is critical since the motivation and rationale is very confusing, and partly due to English issues.

Thank you for your valuable comment and positive feedback. English language corrected, certificate attached. 

2. The introduction hardly leads the reader to the proposed aim and hypotheses. Just to provide an example of what I mean. in the last paragraph of the intro one can read "In the short track, the athletes always

skate to the left, so it was also checked, how much the parameters of the muscles in both lower limbs vary among themselves in the athletes in this asymmetric sport discipline and to determine the reference values for the discipline."...which I believe the authors intended to discuss and point out potential asymmetries in this sport. However, such topic is hardly discussed within the intro and a common reader hardly understands why does it matter (is it really "bad" to be asymmetric in such "a asymmetric sport"). I am not trying to say it does, or does not, matter. But this needs a clear discussion so the reader can get the whole picture of what and why the authors are investigated such thing. Note this is just one example of the lack of a strong rationale within the introduction

Thank you for your valuable suggestions. Much of the introduction and discussion has been improved to better guide the reader.

3. statistical analysis: are you really calculating ICC analysis (note: please report SEM values), correlations, and a 2 factors design with an N =7? I am sorry, but no matter how large the effect size can be...this is potentially tremendously biased. I fully understand this is an elite group, and maybe you only have access to 7 elite athletes. But there are many others statistical approaches to deal with such type of data set (e.g. single-subject analysis.

We fully understand your comment. The reason for calculating ICC and CV is to establish the reliability of the evaluator's measurement in the TMG and the CMJ. In the reviews carried out on the use of TMG, the assessment standard strongly recommends to include the reliability calculation to ensure the quality of the data, since this is a key element (Macgregor et al., 2018; García-García et al., 2019). For this, two assessments were carried out with each athlete about 15 min apart. For TMG, although there were only 7 athletes, data of 8 muscles were collected for each of these seven athletes, which implies a good number of cases for subsequent analysis. For its calculation we have followed the indications of Shrout and Fleiss (1979) and Koo and Li (2016) and for calculating the intra-rater reliability a 2-way mixed-effects is suggested appropriately.

In order to be well explained in the manuscript, we have added the following paragraph at the end of the study design section:

To report the intraday reliability of the measurement (Swift SpeedMat platform and TMG) for each evaluation, two measurements were performed in each athlete separated by a period of 15 min. 

All measures were carried out by the same evaluator who has extensive experience in the use of both tools. Text added as follows:

All measurements were carried out by the same evaluator who had extensive experience in the use of both tools.

In addition, following your suggestion, the SEM has been calculated, which is reflected in the results section:

The reliability values (ICC, 95% CI; CV and %SEM) obtained were: FT .99 (.95-.99), cv 1%, SEM .96%; JH .99 (.94-.99), cv 2.1%, SEM 1.94%; JP .99 (.97-.99), cv 2.1%, 1.87%; Dm .92 (.80-.97), cv 6.5%, SEM 7.35%; Tc .92 (.80 - .96), cv 4.4%, SEM 4.37%; Td .93 (.84 - .97), cv 3.4%, SEM 2.89%.

In the statistical analysis section, the following information has been added:

The percentage of standard error of measurement (%SEM) has also been used as an absolute reliability measure. SEM= √MSE, where MSE is the mean square error term from the repeated-measures ANOVA. %SEM was calculated as SEM/M × 100, where M is the mean of the two intraday measurements.

4. Study design: At a certain point I got very confused. The authors mentioned a pre vs post, left vs right, and experimental vs control (note: this last one does not seem the best terminology to be). I couldn't see this design being tested in your statistical approach, I may have missed some.

Thank you for your valuable comment. The study design has been corrected. In fact, the statement regarding groups was used in an unfortunate way. As a result, the control group was changed throughout the manuscript to a test after recovery training and the experimental group to a test after strength training.

---

## [Decision Letter · Decision Letter 1]

29 Nov 2021

Asymmetry and changes in the neuromuscular profile of short-track athletes as a result of strength training

PONE-D-21-22288R1

Dear Dr. Pakosz,

We’re pleased to inform you that your manuscript has been judged scientifically suitable for publication and will be formally accepted for publication once it meets all outstanding technical requirements.

Kind regards,

Emiliano Cè

Academic Editor

PLOS ONE

Additional Editor Comments (optional):

Reviewers' comments:

Reviewer's Responses to Questions

**Comments to the Author**

1. If the authors have adequately addressed your comments raised in a previous round of review and you feel that this manuscript is now acceptable for publication, you may indicate that here to bypass the “Comments to the Author” section, enter your conflict of interest statement in the “Confidential to Editor” section, and submit your "Accept" recommendation.

Reviewer #1: All comments have been addressed

2. Is the manuscript technically sound, and do the data support the conclusions?

Reviewer #1: Yes

3. Has the statistical analysis been performed appropriately and rigorously? 

Reviewer #1: Yes

4. Have the authors made all data underlying the findings in their manuscript fully available?

Reviewer #1: Yes

5. Is the manuscript presented in an intelligible fashion and written in standard English?

Reviewer #1: Yes

6. Review Comments to the Author

Reviewer #1: (No Response)

7. PLOS authors have the option to publish the peer review history of their article (what does this mean?). If published, this will include your full peer review and any attached files.

Reviewer #1: No

---

## [Editor Report · Acceptance letter]

9 Dec 2021

PONE-D-21-22288R1 

Asymmetry and changes in the neuromuscular profile of short-track athletes as a result of strength training 

Dear Dr. Pakosz:

I'm pleased to inform you that your manuscript has been deemed suitable for publication in PLOS ONE. Congratulations! Your manuscript is now with our production department. 

Kind regards, 

on behalf of

Professor Emiliano Cè 

Academic Editor

PLOS ONE